# Electrostatic control of the proximity effect in the bulk of semiconductor-superconductor hybrids

Nick van Loo [1,3], Grzegorz P. Mazur [1,3] ✉, Tom Dvir [1], Guanzhong Wang[1], Robin C. Dekker [1], Ji-Yin Wang[1], Mathilde Lemang[1], Cristina Sfiligoj[1], Alberto Bordin[1], David van Driel[1], Ghada Badawy[2], Sasa Gazibegovic[2], Erik P. A. M. Bakkers[2] & Leo P. Kouwenhoven[1] ✉

The proximity effect in semiconductor-superconductor nanowires is expected to generate an induced gap in the semiconductor. The magnitude of this induced gap, together with the semiconductor properties like spin-orbit coupling and $g$-factor, depends on the coupling between the materials. It is predicted that this coupling can be adjusted through the use of electric fields. We study this phenomenon in InSb/Al/Pt hybrids using nonlocal spectroscopy. We show that these hybrids can be tuned such that the semiconductor and superconductor are strongly coupled. In this case, the induced gap is similar to the superconducting gap in the Al/Pt shell and closes only at high magnetic fields. In contrast, the coupling can be suppressed which leads to a strong reduction of the induced gap and critical magnetic field. At the crossover between the strong-coupling and weak-coupling regimes, we observe the closing and reopening of the induced gap in the bulk of a nanowire. Contrary to expectations, it is not accompanied by the formation of zero-bias peaks in the local conductance spectra. As a result, this cannot be attributed conclusively to the anticipated topological phase transition and we discuss possible alternative explanations.

When a semiconductor is coupled to a superconductor, the resulting hybrid is expected to inherit properties of both. The combination of these properties can be exploited to create exotic phases of matter[1,2]. For example, a magnetic field can trigger the transition to a phase of topological superconductivity in semiconducting nanowires with strong spin–orbit coupling[3,4]. In theory, this should be accompanied by the formation of Majorana zero modes (MZMs) at the ends, together with a closing and reopening of the superconducting gap in the bulk of the hybrid[5,6]. In general, the proximity effect induces superconductivity in the semiconductor as a result of Andreev reflection at the interface between the materials. This effect manifests itself as the emergence of an induced superconducting gap $\Delta_i$ in the semiconductor. The size of this gap depends on the size of the proximitizing superconductor $\Delta_{SC}$, as well as the coupling between the materials[7]. Importantly, the coupling also affects various other properties of the hybrid, such as the spin–orbit coupling and $g$-factor. Moreover, it is expected to be tunable through the use of electric fields[8,9].

In experiments, a modest tunability of the superconducting gap[10,11] and the $g$-factor of Andreev bound states (ABSs)[12,13] have been reported. However, most experiments to date rely on tunneling measurements at the end of a nanowire, which only provide information on

[1]QuTech and Kavli Institute of Nanoscience, Delft University of Technology, 2628 CJ Delft, The Netherlands. [2]Department of Applied Physics, Eindhoven University of Technology, 5600 MB Eindhoven, The Netherlands. [3]These authors contributed equally: Nick van Loo, Grzegorz P. Mazur. ✉e-mail: g.p.mazur@tudelft.nl; l.p.kouwenhoven@tudelft.nl

the local density of states. Yet, it remains unknown what information these observations provide about the proximity effect in the bulk of a hybrid. Advances in nanofabrication now enable the study of semiconductor-superconductor hybrids in a three-terminal geometry[14–16]. In addition to the local density of states at the two ends of a nanowire, such devices allow the nonlocal conductance to be measured. Nonlocal transport is fundamentally carried by states in the nanowire that couple to both leads. Moreover, it requires their energy to reside in an energy window between the gap of the superconductor and the induced gap in the semiconductor[5], and thus can be used to directly determine the induced gap in the bulk of the hybrid[17]. Measurements in this geometry have been used to observe the closing of the induced gap[18], map the local charge of ABSs[19,20], investigate the quasiparticle wavefunction composition[21] and search for topological superconductivity in a variety of platforms[22,23].

In this article, we investigate the effect of gate-induced electric fields on the bulk of InSb nanowires, proximitized by Al/Pt films[24]. To do this, we utilize nonlocal spectroscopy. We demonstrate that the devices can be tuned into a strongly-coupled regime with an induced gap close to that of the Al/Pt shell. Likewise, gate voltages can be used to significantly reduce the induced gap and eventually fully close it. By applying a parallel magnetic field, we show that wires in the strong-coupling regime can have critical magnetic fields close to that of the superconducting shell. On the other hand, a gate-reduced coupling drastically lowers the critical field.

The three-terminal devices presented in this work are fabricated using our shadow-wall lithography technique[15,16]. In Fig. 1a we depict the device schematic of a nanowire hybrid used in these experiments. A set of pre-patterned bottom gates is separated from the InSb nanowire by a thin layer of HfO$_2$. Voltages on the two tunnel gates, $V_{TL}$ and $V_{TR}$, are used to induce tunnel barriers in the exposed semiconducting segments. The super gate voltage $V_{SG}$ is used to apply an electric field in the bulk of the hybrid. The nanowire is covered on three facets by an Al/Pt film, where the Pt serves to enhance the critical magnetic field of the Al film[24]. This superconducting shell extends onto the substrate, forming the connection to ground. Two Cr/Au contacts are fabricated at the ends of the wire. The devices are measured by individually applying bias voltages, $V_L$ and $V_R$, to the left and right leads. The conductance matrix is obtained by measuring the differential conductances $g_{ij} \equiv dI_i/dV_j$, with i, j = L, R using standard lock-in techniques (see the Methods section for details of device fabrication and measurement).

In Fig. 1b, we illustrate the expected effect of electric fields on the bulk of the hybrid as calculated by[25]. For negative gate voltages (Fig. 1b(i)), electrons accumulate near the semiconductor-superconductor interface which results in a strong coupling to the superconductor. As a consequence, the semiconducting properties of the hybrid are strongly renormalized. We refer to this as the strong-coupling regime in the rest of this work. On the other hand, electrons can accumulate far from the interface through the application of positive gate voltages (Fig. 1b(iii)). This results in a diminished coupling with unproximitized states in the hybrid, to which we refer as the weak-coupling regime. Finally, there is a crossover between these two regimes (Fig. 1b(ii)) where electrons still maintain superconducting correlations, while their semiconducting properties are only moderately renormalized. As a result, this crossover is expected to be optimal for the emergence of topological superconductivity[25]. Furthermore, the application of an electric field also changes the electron density in the hybrid. Due to quantum confinement we expect the formation of discrete subbands, each with their own coupling strength. Thus, applied gate voltages should be able to tune the hybrid between the different subbands.

To characterize the different coupling regimes, we determine the induced gap in our devices using nonlocal spectroscopy. The transport mechanisms involved in such measurements are schematically

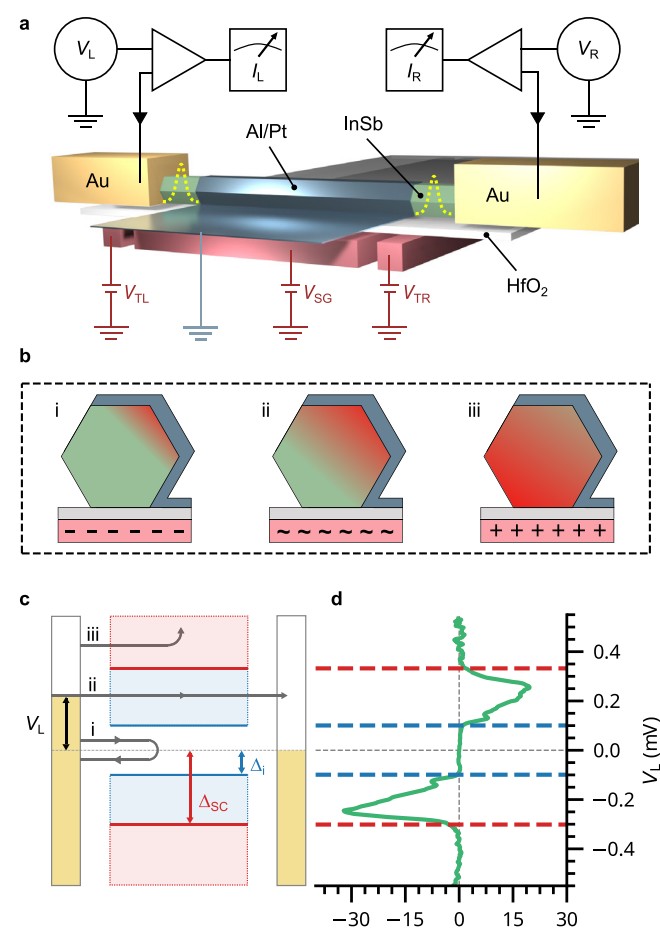

**Fig. 1 | Illustration of the experimental setup and nonlocal measurement technique. a** Schematic of a three-terminal hybrid device and the measurement circuit. The superconducting shell is grounded through its connection to the film on the substrate. Yellow dashed potentials indicate the formation of tunnel barriers in the semiconducting junctions. **b** Illustration of three different coupling regimes between the superconductor and semiconductor. (i) Strong-coupling: electrons (red) are confined at the interface which results in a renormalization of the semiconducting properties. (ii) Crossover: predicted to be optimal for the formation of a topological superconductor. (iii): Weak-coupling: electrons accumulate far from the interface, which can result in unproximitized states. **c** Transport schematic of nonlocal measurements. (i) Below $\Delta_i$ (blue), only local processes are possible. (ii) In between $\Delta_i$ and $\Delta_{SC}$, nonlocal transport can occur. (iii) Above $\Delta_{SC}$ (red), electrons are drained to ground. **d** Example of measured nonlocal conductance $g_{RL}$ taken on device A, corresponding to the diagram in (**c**). Blue and red dashed lines indicate the induced and superconductor gap, respectively.

depicted in Fig. 1c, together with an example of the resulting nonlocal conductance $g_{RL}$ in Fig. 1d. If the applied bias $V_L$ is below the induced gap $\Delta_i$, electrons from the lead can only enter the superconducting region through Andreev reflection (Fig. 1c(i)). This results in the formation of Cooper pairs, which drain away into the superconducting lead. As a consequence, no nonlocal conductance is observed below the induced gap (Fig. 1d). Similarly, electrons injected above the gap of the superconductor $\Delta_{SC}$ are likely to drain to the ground without reaching the other side[26] (Fig. 1c(iii)). However, if the applied bias is larger than $\Delta_i$ but below $\Delta_{SC}$, injected electrons can reach the opposite lead of the device. This results in a finite nonlocal conductance as shown in Fig. 1d, from which $\Delta_i$ (dashed blue lines) and $\Delta_{SC}$ (dashed red lines) can be estimated. In the Methods section and Supplementary section I, we describe how these parameters are determined from the data. While this picture helps to understand three-terminal

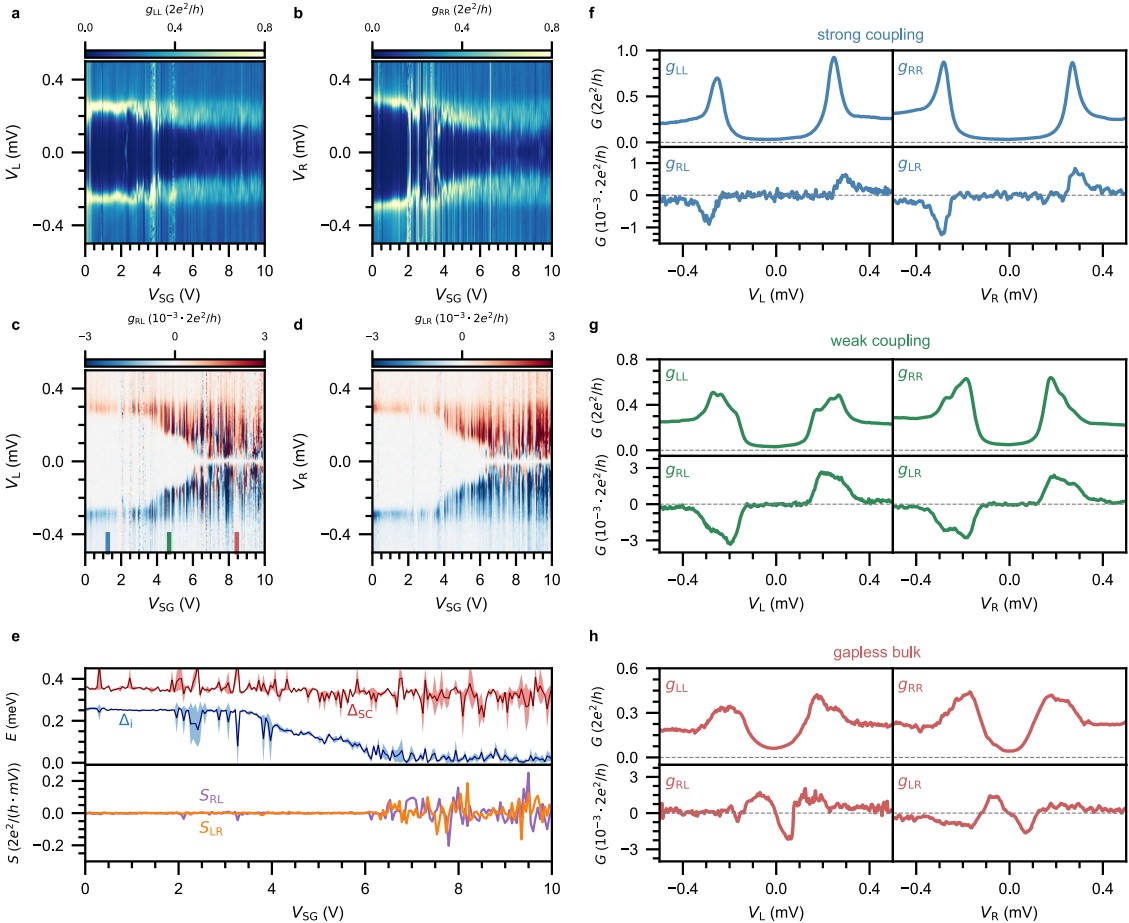

**Fig. 2 | Tunability of the induced superconducting gap through electrostatic gating. a**–**d** Conductance matrix measured as a function of $V_{SG}$ in the absence of a magnetic field on device B (8 $\mu$m long hybrid). At low $V_{SG}$, a large induced gap is observed in $g_{RL}$ and $g_{LR}$ (panels **c**, **d**). For increasing $V_{SG}$, the induced gap gradually decreases and eventually fully closes. At the same time, the superconducting gap in $g_{LL}$ and $g_{RR}$ (panels **a**, **b**) becomes soft. **e** Top: $\Delta_i$ (blue) and $\Delta_{SC}$ (red). Dark colors represent the mean of four values, obtained from the positive and negative biases of the two nonlocal signals. Similarly, the shaded areas correspond to the standard deviation. Bottom: calculated nonlocal slope at zero bias for $g_{RL}$ (purple) and $g_{LR}$ (orange). **f** Linecuts of the conductance matrix taken at $V_{SG} = 1.26$ V in the strong-coupling regime, where a large induced gap is observed. **g** Linecuts of the conductance matrix taken at $V_{SG} = 4.67$ V in the weak-coupling regime, where the induced gap is significantly reduced. **h** Linecuts of the conductance matrix taken at $V_{SG} = 8.44$ V. The induced gap is closed as visible in $g_{RL}$ and $g_{LR}$, whereas the superconducting gap in $g_{LL}$ and $g_{RR}$ has turned soft.

measurements, we note that nonlocal processes can involve energy relaxation of the injected electrons as well as non-equilibrium effects not captured by the single-particle transport theory[21]. We further elaborate on this in Supplementary section II.D.

## Results and discussion

First, we investigate the gate tunability of the induced gap. We measure the full conductance matrix of a device as a function of super gate voltage $V_{SG}$ at zero magnetic field. In Fig. 2, we show such a measurement on a long nanowire hybrid (device B, 8 $\mu$m long). Panels c and d depict the nonlocal signals $g_{RL}$ and $g_{LR}$, in which the induced gap is directly visible as the white area between two anti-symmetric peaks. For a large range of low gate voltages, the peaks in the nonlocal signal are relatively sharp. This indicates that the difference between $\Delta_i$ and $\Delta_{SC}$ is small, and so it is associated with the strong-coupling regime. Above a certain gate voltage $V_{SG} > 3.5$ V, the peaks gradually become wider. This signals the reduction of the induced gap, as the coupling between the semiconductor and superconductor is decreased. The induced gap fully closes above $V_{SG} > 6$ V, which means that at this point there reside states in the bulk of the nanowire which do not couple at all to the superconductor. To better visualize the effect of $V_{SG}$, in the top panel of Fig. 2e we plot the behavior of the induced gap in the bulk (blue) and the gap of the superconducting shell (red). In addition, we

show in the bottom panel the nonlocal slope at zero bias[18]. This parameter is defined as $S_{ij} \equiv d^2 I_i/dV_j^2|_{V_j = 0}$, with $S_{RL}$ presented in purple and $S_{LR}$ in orange. Indeed, their deviation from zero above $V_{SG} > 6$ V confirms that the hybrid has become gapless. We generically observe the tunability of the induced gap, and hence the coupling between the semiconductor and superconductor. However, the application of an electric field does not exclusively tune the coupling but also controls the density in the hybrid. Typically, we observe a sudden onset of the reduction of $\Delta_i$ while the magnitude of the nonlocal signal increases concurrently. This behavior has theoretically been related to the occupation of an additional subband with a reduced coupling[8]. Still, it remains unknown how many sub-bands are active in our hybrids.

It is particularly interesting how the reduction of the induced gap is also reflected in the local signals $g_{LL}$ and $g_{RR}$, which are displayed in Fig. 2a, b. In the strong-coupling regime, the local signals exhibit two sharp coherence peaks and for the majority of the gate voltages, a clean superconducting gap. However, some states can be seen in these spectra which do not correlate between the two panels nor show up in the nonlocal signals—a confirmation that these states are confined to the local tunnel junctions. Exemplary linecuts in this regime of the full conductance matrix are shown in Fig. 2f. In $g_{LL}$ and $g_{RR}$, we see a typical local spectrum which in literature is referred to as a hard superconducting gap. While the sub-gap conductance does not actually go

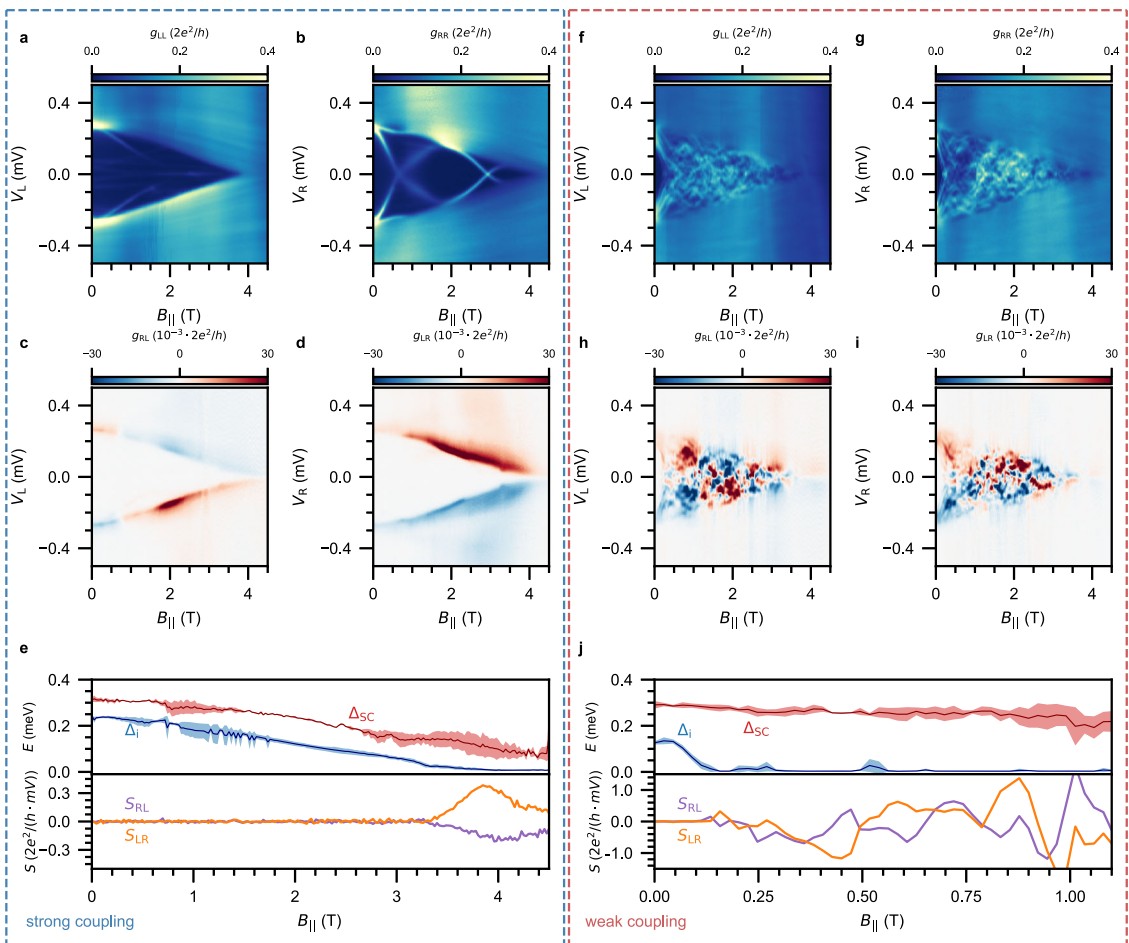

**Fig. 3 | Magnetic-field evolution of the induced superconducting gap.**
**a**–**d** Conductance matrix measured as a function of $B_\parallel$ for device C (1 $\mu$m hybrid) in the strong-coupling regime ($V_{SG} = -0.75$ V), where the induced gap only closes at large magnetic fields as visible in $g_{RL}$ and $g_{LR}$ (panels **c**, **d**). In $g_{LL}$ and $g_{RR}$ (panels **a**, **b**), a few sub-gap states localized in the vicinity of the tunnel junctions are observed. **e** Top panel: $\Delta_i$ (blue) and $\Delta_{SC}$ (red) in the strong-coupling regime.

Bottom panel: Nonlocal slope extracted from $g_{RL}$ (purple) and $g_{LR}$ (orange).
**f**–**i** Conductance matrix measured as a function of $B_\parallel$ for device C (1 $\mu$m hybrid) in the weak-coupling regime ($V_{SG} = 0.5$ V), where the induced gap instead closes at small magnetic fields as visible in $g_{RL}$ and $g_{LR}$ (panels **h**, **i**). This is also reflected in $g_{LL}$ and $g_{RR}$ (panels **f**, **g**). **j** Top panel: $\Delta_i$ (blue) and $\Delta_{SC}$ (red) in the strong-coupling regime. Bottom panel: Nonlocal slope extracted from $g_{RL}$ (purple) and $g_{LR}$ (orange).

to zero, we note that the junctions are relatively transparent. This results in a significant amount of Andreev reflection[27], which contributes only to the local conductance. To confirm this, we have repeated similar measurements in the tunneling regime (see Supplementary information Fig. S9). Indeed, the hard gap is also visible in $g_{RL}$ and $g_{LR}$, which show zero response outside of the two anti-symmetric peaks. As $V_{SG}$ increases, the semiconductor-superconductor coupling is reduced. Linecuts in the regime of weak coupling are shown in Fig. 2g. The coherence peaks visible in $g_{LL}$ and $g_{RR}$ have now broadened significantly, while the sub-gap conductance still only contains contributions from Andreev reflection. The peaks in $g_{RL}$ and $g_{LR}$ have also broadened accordingly, while the absence of signal in between still evidences a hard gap. This changes when $V_{SG}$ is increased further, as show in Fig. 2h. The absence of a flat part in $g_{RL}$ and $g_{LR}$ now indicates that an induced gap is absent in the system. This is also seen in $g_{LL}$ and $g_{RR}$, where the conductance close to zero bias is now increased beyond what can be explained by Andreev reflection. Indeed, the nanowire now exhibits a soft gap as measured from the local spectra, while the nonlocal signals demonstrate that the hybrid is gapless. Such a soft gap has long been attributed only to the quality of the semiconductor-superconductor interface[28]. Yet, here we show that this is not the full story: a soft gap can equally well exist in hybrids with a pristine interface. In this case, it is caused by a combination of a weak semiconductor-superconductor coupling and increasing electron

density in the nanowire. Since a high electron density and weak or absent semiconductor-superconductor coupling are unfavorable conditions for the formation of a topological superconducting phase, this demonstrates that hybrids with a soft gap in the local spectra are unlikely to undergo a topological phase transition. Moreover, topological superconductivity has to be realized by a Zeeman-driven gap closing and reopening, which is not possible to realize in devices gapless already at zero field.

We proceed by exploring the effect of parallel magnetic fields $B_\parallel$ on the induced gap of a 1 $\mu$m long hybrid (device C). In Fig. 3, we present two field sweeps of the nanowire in the two extreme regimes. In the strong-coupling regime (Fig. 3a–e), the induced gap decreases slowly with magnetic field. For $g_{RL}$ and $g_{LR}$ it can be seen that the white area between the anti-symmetric peaks persists up to almost $B_\parallel = 4$ T. At the same time, a few states can be observed in $g_{LL}$ and $g_{RR}$. Again, the absence of correlation between the two sides and the absence of these states from the nonlocal signals confirms they are confined locally near the tunnel junctions. By looking at the estimated induced gap and the nonlocal slope in Fig. 3e, we observe an induced critical field $B_\parallel^c = 3.5$ T. The outer ridge of the nonlocal signal decreases more slowly, which indicates that the shell maintains a superconducting gap (red) up to higher fields. By fitting the linear part of the induced-gap closing to the Zeeman energy $E_Z = g\mu_B B/2$, we estimate the g-factor to be $g = 2.3$ (see Supplementary section II). This demonstrates that the semiconductor

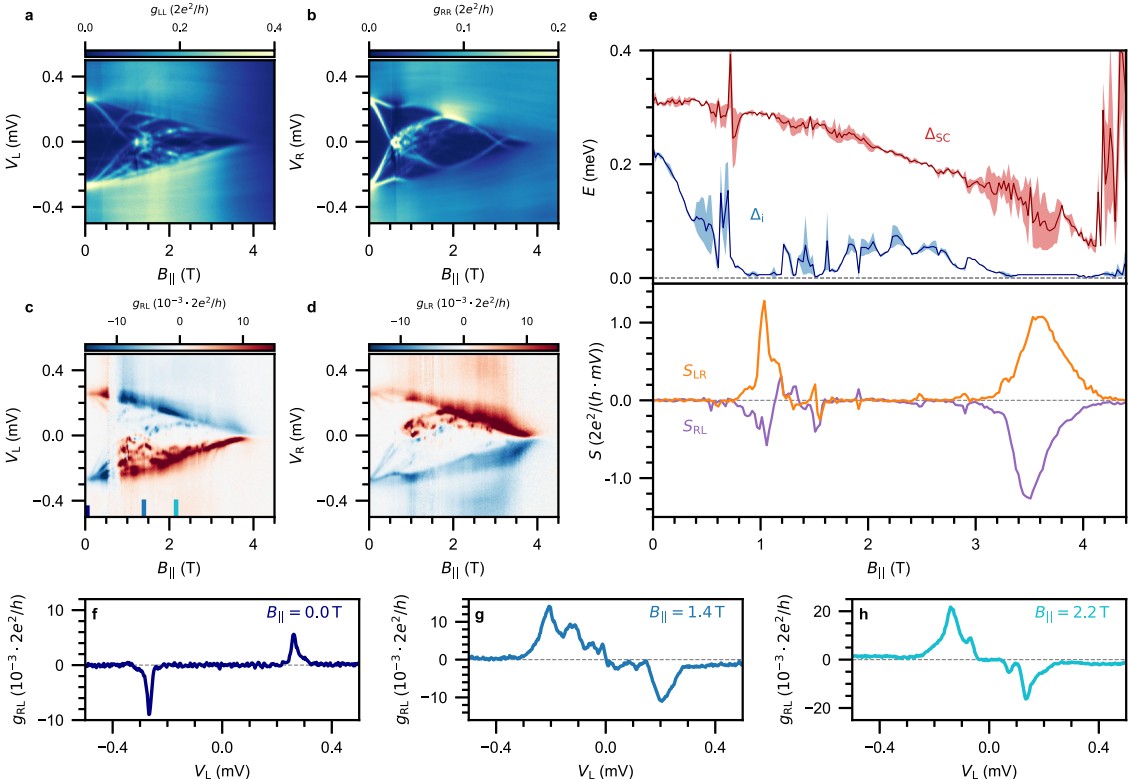

**Fig. 4 | The closing and reopening of the induced superconducting gap.**
**a–d** Conductance matrix measured as a function of $B_\parallel$ at $V_{SG} = -0.3$ V for device C (1 $\mu$m hybrid). A closing and reopening of the induced gap is observed in $g_{RL}$ and $g_{LR}$ (panels **c**, **d**), but it is not accompanied by the formation of zero-bias peaks in $g_{LL}$ and $g_{RR}$ (panels **a**, **b**). **e** $\Delta_i$ (blue) and $\Delta_{SC}$ (red) corresponding to the conductance matrix in (**a–d**). Bottom: Nonlocal slope extracted from $g_{RL}$ (purple) and $g_{LR}$ (orange). **f–h** Nonlocal conductance $g_{RL}$ presented for (**f**) $B_\parallel = 0$ T with a large induced gap, (**g**) $B_\parallel = 1.4$ T illustrating a closed induced gap, (**h**) $B_\parallel = 2.2$ T showing flat nonlocal conductance around zero-bias corresponding to a reopening of the induced gap.

properties are indeed strongly renormalized in this regime[8,9]. Such a low $g$-factor and the absence of any states below $\Delta_i$ may suggest that the semiconductor is depleted. Yet, we observe that the induced critical field in the strong-coupling regime varies significantly from wire to wire, and likely depends on the microscopic details (see Supplementary section II). Moreover, we note that the addition of Pt in the shell causes its $g$-factor to be reduced close to zero, so that the effective $g$-factor in the hybrid can be reduced below $g = 2$[24]. In the weak-coupling regime (Fig. 3f–j) on the contrary, the induced gap closes quickly upon the application of the magnetic field. Thereafter, the spectrum remains gapless and filled with a plethora of states. This is also reflected in $g_{LL}$ and $g_{RR}$, where the same states are visible. From both the induced gap and the nonlocal slope in Fig. 3j, we observe an induced critical field $B_\parallel^c = 0.16$ T. We estimate a $g$-factor of $g = 54$, although this value can be overestimated as orbital effects of the magnetic field are more prominent in this regime[29–31]. The rapid closing of the induced gap confirms that the hybrid inherits more of the semiconductor properties in the weak-coupling regime.

We next turn our attention to the crossover between these two regimes, which is expected to be optimal for the formation of a topological superconducting phase[25]. In Fig. 4a–d, the conductance matrix of the same nanowire (device C, 1 $\mu$m) taken at $V_{SG} = -0.3$ V is shown. In the nonlocal spectra (Fig. 4c, d), we see a collection of states moving down in energy as the magnetic field is increased. The induced gap closes around $B_\parallel^c = 0.8$ T and reopens around $B_\parallel^c = 1.6$ T. The induced gap (blue) and nonlocal slope are presented in Fig. 4e. Here, the closing and reopening of the induced gap is directly visible. The reopened gap reaches energies of $\Delta_i = 50 \mu$eV, which is similar to predictions of the gap size in topological systems[6]. The reopening is also reflected in the behavior of the nonlocal slope, which deviates from

zero around $B_\parallel = 1$ T before returning to zero again at higher fields. Figure 4f-h provide linecuts from $g_{RL}$, emphasizing that the induced gap is finite at zero field, closed at intermediate field, and reopened at higher fields. However, neither of the local signals (Fig. 4a and b) exhibit zero-bias peaks. This suggests that the observed feature does not originate from a topological phase with Majorana zero modes at the ends, extended over the full length of the hybrid. Yet, it may be possible that the presence of tunnel gates generates a smooth potential profile near the ends of the wire. In this case, the local spectra only represent the presence of bound states formed on the smooth potential, while pushing the Majorana zero modes towards the center of the hybrid—effectively decoupling them from the leads[32,33]. Similar effects are expected to be caused by the device disorder independent of the tunnel gate voltage[34]. Accordingly, the gap reopening in the bulk should remain visible in the nonlocal spectra as this effectively measures the largest gap in the system, while no zero-bias peaks are observed in the local signals. This scenario is supported by the observation that the local signals $g_{LL}$ and $g_{RR}$ do not appear to depend on the length of the hybrid and are not always correlated, as we elaborate on Supplementary section II. On the contrary, it is also possible that the reopening of the gap has a topologically trivial origin. The hybrid segment of this device is only 1 $\mu$m long, so that it is likely to be within the short wire limit. In this case, the resulting spectrum is comprised of discrete energy levels with a small energy spacing. Both the Zeeman and orbital contributions of the magnetic field allow these states to come down and cross zero energy. However, in this limit there is no band structure forming in the nanowire, making the concept of topology ill-defined[35]. Alternatively, the observed gap reopening can originate from two sets of trivial ABSs localized near the nanowire junctions. In this case, spatial overlap due to a long localization length

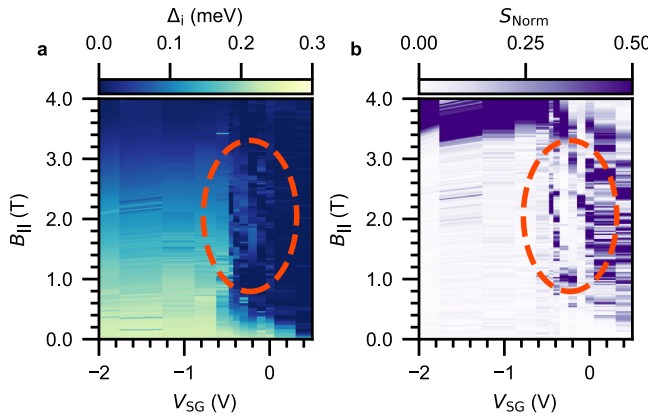

**Fig. 5 | Phase diagrams of the induced superconducting gap and the nonlocal slope. a** Induced gap as a function of $V_{SG}$ and $B_{\parallel}$. **b** Normalized nonlocal slope $S_{Norm}$ as a function of $V_{SG}$ and $B_{\parallel}$. Dashed orange ellipses highlight the reopening of the induced gap, which occurs in a small but finite range of $V_{SG}$ values. Data taken on device C (1 $\mu$m long hybrid).

can enable transport through the hybrid[36]. Likewise, such states can cross zero energy without invoking a topological phase transition.

Finally, to enhance the picture we map out the induced gap of a nanowire as a function of parallel magnetic field and super gate voltage. In Fig. 5a, we present such an induced gap diagram for the same 1 $\mu$m long hybrid (device C). To complement this diagram, we show the corresponding normalized nonlocal slope $S_{Norm}$ at zero bias in Fig. 5b. This quantity captures the collective behavior of the nonlocal slope from the two nonlocal signals, remaining close to zero whenever an induced gap is present in the hybrid. It is defined as $S_{Norm} = |S_{RL}S_{LR}|/\sqrt{|S_{RL}S_{LR}|}$ where the normalization is done independently for every gate voltage. In the strong-coupling regime below $V_{SG} < -0.5\,V$, we see that $\Delta_i$ decays slowly when the magnetic field is increased. It closes around $B_{\parallel}^c = 3.5$ T, which is also reflected in $S_{Norm}$ as it deviates from zero. In contrast, above $V_{SG} > -0.1\,V$ the semiconductor-superconductor coupling is strongly diminished which results in a significant reduction of $\Delta_i$ and $B_{\parallel}^c$. Near the crossover between $-0.4 < V_{SG} < -0.1\,V$ as indicated by the dashed orange ellipses, the closing of $\Delta_i$ is followed by its reopening at higher magnetic fields. This is also visible in the behavior of $S_{Norm}$, which becomes finite when the gap closes and returns to zero at the reopening. Importantly, the reopening occurs in a finite but narrow range of gate voltages. While a strong reduction of $B_{\parallel}^c$ is generically observed in our hybrids, only one out of the eleven nanowires studied in detail showed a subsequent reopening of the induced gap. In Supplementary section II, we show phase diagrams and representative overviews of additional nanowires studied in this work.

In conclusion, we have demonstrated that electric fields can be used to control the bulk properties of InSb nanowires proximitized by Al/Pt films, using a three-terminal geometry. Even though this has been attempted in the past using two-terminal experiments[10,12], local tunneling spectroscopy does not allow for discriminating between states localized in the junction area which are known to possess a gate-tunable coupling[37]. Such states are present in virtually all our local spectra, which demonstrates that nonlocal measurements are truly necessary to properly investigate the bulk properties of semiconductor-superconductor hybrids. On the one hand, a strong-coupling regime can be achieved where the induced gap is large and closes only at high magnetic fields. This corresponds to a metallized nanowire which has a strong renormalization of the semiconducting properties. In contrast, a weak-coupling regime can be realized where the induced gap and critical field are strongly reduced. In particular,

the induced gap can be fully closed at zero magnetic field. We have demonstrated that under these conditions the system also exhibits a soft gap in the local spectra of the nanowire. Thus, the presence of a soft gap is a clear indication that the bulk of the hybrid posesses a high electron density with a weak (or absent) semiconductor-superconductor coupling. Indeed, the strong-coupling regime acts too much like a regular superconductor whereas the weak-coupling regime acts too much like an ordinary semiconductor. Neither of these regimes have an interesting combination of both properties - making a large part of the gate voltage range irrelevant for the realization of a topological superconductor. Only at the crossover one can hope to find the right combination of semiconductor and superconductor properties.

By mapping out the induced gap diagram of a 1 $\mu$m nanowire near the crossover, we do observe a closing and reopening of the induced gap in a finite range of magnetic fields and gate voltages. However, the corresponding local signals reveal an absence of zero-bias peaks. As a consequence, the gap reopening cannot be conclusively attributed to the existence of a topological phase. We speculate that the density in the hybrids is too high whenever the coupling is weakened[8]. In fact, it is currently unclear what are the optimal density and coupling for reaching a topological phase in InSb/Al based hybrids. Thus, a desireable future improvement would be to decouple the semiconductor and superconductor via an epitaxial barrier, such that density in the wire and the coupling could be tuned independently[38].

## Methods
### Device fabrication
The nanowire hybrids presented in this work are fabricated on pre-patterned substrates, following the shadow-wall lithography technique described in refs. 15, 16. Intrinsic silicon wafers (2 $k\Omega$ cm) with 285 nm thermal SiO$_x$ serve as the base for the device substrates. Local bottom gates are patterned with standard electron-beam lithography (EBL) techniques, using PMMA 950k A2 spun at 4 krpm for one minute followed by 10 minutes of baking on a 185 °C hot plate. After development of the resist using a 3:1 solution of IPA and MIBK, 3 nm Ti and 17 nm Pd are deposited as the gate metal using e-beam evaporation at 0.5 Å/s and 1 Å/s, respectively. Subsequently, bond pads are patterned with EBL using PMMA 950k A6 spun at 4 krpm for one minute and hot-baked at 185 °C for 10 min. After development, 50 nm of W is sputtered using RF-sputtering at 150 W in an Ar pressure of 20 $\mu$bar. Next, the substrates are covered with 17 nm high-quality HfO$_x$ gate dielectric grown at 110 °C using atomic layer deposition (ALD). Finally, shadow walls are patterned on top of the dielectric. FOx-25 (HSQ) is spun at 1.5 krpm for one minute, followed by 2 minutes of hot baking at 180 °C and patterning with EBL. The HSQ is then developed with MF-321 at 60 °C for 5 minutes and the substrates are subsequently dried using critical point dryer (CPD).

The InSb nanowires are grown using metalorganic vapor-phase epitaxy, as described in ref. 39. The nanowires are placed on top of the gates using an optical nanomanipulator setup. Samples are placed in a custom e-beam evaporator, where the native nanowire oxide is removed and the superconductor is deposited. To obtain a pristine, oxide-free semiconductor surface, a gentle oxygen removal is accomplished via atomic hydrogen radical cleaning. For this purpose, a custom-made H radical generator is installed in the load lock of an aluminum electron-gun evaporator. It consists of a gas inlet for H$_2$ molecules connected to a mass-flow controller and a tungsten filament at a temperature of about 1700 °C that dissociates a fraction of the molecules into hydrogen radicals. The optimal removal of the native oxide is achieved for a process duration of 60 mins and at a H$_2$ pressure of $6.3 \times 10^{-5}$ mbar. This recipe, which is used for all the devices shown in this paper, results in a constant EDX count of oxygen at the interface as shown in the previous works utilizing our shadow wall

lithography technique (see Refs. [15],[16]). After the native oxide removal, the samples are cooled down to 138 K and thermalized for one hour. The Al is then deposited with a rate of 0.05 Å/s, alternating 2 nm at 15° and 45° angles with respect to the substrate for a total of 8 nm. Subsequently, a 2 Å Pt layer is deposited at 30° following the approach of[24]. Finally, the samples are capped with evaporated $AlO_x$ (-0.2 Å/s) to prevent oxidation of the shell-substrate connection.

Ohmic contacts are fabricated ex-situ after the superconductor deposition. PMMA 950k A6 is spun at 4 krpm and subsequently cured at room temperature in a vacuum oven to prevent intermixing at the pristine InSb-Al interface. Contacts are patterned using EBL, and Ar milling is used to remove the native oxide prior to deposition of 10 nm Cr and 120 nm Au using e-beam evaporation at 0.5 Å/s and 1.5 Å/s, respectively.

## Measurement details

Transport measurements are conducted in dilution refrigerators with a base temperature of ~20 mK. All magnetic field measurements presented in this work have the magnetic field aligned parallel to the nanowire using 3-axis vector magnets. We have used two adaptations of the three-terminal circuit presented in[40], which are shown in supplementary Fig. 5. For the measurements on device C, the Al/Pt film is grounded at room temperature, so that there is a finite resistance originating from the fridge line and filters in between the ground and the sample. This can give rise to voltage-divider effects, which we correct. We do this using formula (14) of ref. [40]:

$$G(\mathbf{V}) = G'(\mathbf{V}^{\text{applied}})(\mathbb{I} - ZG'(\mathbf{V}^{\text{applied}}))^{-1} \qquad (1)$$

Here, $G(\mathbf{V})$ is the corrected conductance matrix, $G'(\mathbf{V}^{\text{applied}})$ is the measured conductance matrix, $\mathbf{V} = (V_L, V_R)^T$ is the bias on the sample, $\mathbf{V}^{\text{applied}} = (V_L^{\text{applied}}, V_R^{\text{applied}})^T$ is the bias applied by the voltage sources and $Z$ is the impedance matrix:

$$Z = \begin{bmatrix} R_L + R_g & R_g \\ R_g & R_R + R_g \end{bmatrix} \qquad (2)$$

For device C, the circuit uses a grounding line resistance $R_g = 2834\,\Omega$ and biasing line resistances $R_L = R_R = 3944\,\Omega$. In addition, the DC voltage drop on the sample is corrected by measuring the current $\mathbf{I} = (I_L, I_R)^T$ using

$$\mathbf{V} = \mathbf{V}^{\text{applied}} - Z\,\mathbf{I} \qquad (3)$$

The other samples in this work have used an adapted circuit, where the Al/Pt film is directly connected to the cold ground in the fridge as well as the ground at room temperature. The drawback is that the thermal voltage $V_{\text{th}}$ introduces an offset in the bias voltages applied to the left and right lead, which needs to be corrected. The connection of the Al/Pt film to ground at room temperature enables the direct measurement of this thermal voltage. Yet, this largely eliminates voltage-divider effects from the measurements. The DC voltage drop is also corrected using equation (3) with $R_g = 0\,\Omega$. By maintaining a low conductance compared to the line resistances, we can use the simplified correction of formula (16) of ref. [40]:

$$G(\mathbf{V}) = G'(\mathbf{V}^{\text{applied}}) + \begin{bmatrix} R_L\,g_{LL}^2 & 0 \\ 0 & R_R\,g_{RR}^2 \end{bmatrix} \qquad (4)$$

Note that this simplification can give an error in the nonlocal signals when the zero-bias conductance in the receiving junction is large, either due to the presence of a sub-gap state or Andreev reflection. This leads to a bias-independent offset, which in some measurements we correct by subtracting the offset as calculated from sub-gap or finite-bias conductance values.

When measuring the conductance matrix, the bias on the left contact of the device $V_L$ is swept first while the bias on the other side $V_R$ is set to zero. Before sweeping the bias, the thermal voltage is measured and the bias offsets is calibrated accordingly. The corresponding matrix elements $g_{LL}$ and $g_{RL}$ are recorded. Next, the right-contact bias $V_R$ is swept while setting the bias on the left contact $V_L$ to zero and the remaining two conductance matrix elements $g_{RR}$ and $g_{LR}$ are recorded. For the super gate sweeps presented in this work, we aim to maintain a constant transmission in both the nanowire junctions. To do this, the two tunnel gate voltages $V_{TL}$ and $V_{TR}$ are automatically adjusted each time the super gate voltage is changed. This is done by looking at the out-of-gap local conductances $g_{LL}$ and $g_{RR}$. If one of the conductances is found to deviate more than $0.005 \times 2e^2/h$ from the specified value, the respective tunnel gate voltage is tuned to bring the out-of-gap local conductance back to the specified value.

## Data analysis

We extract the induced gap $\Delta_i$ and the gap of the superconducting film $\Delta_{SC}$ from the nonlocal spectra $g_{RL}$ and $g_{LR}$, as a function of various device parameters like the super gate voltage $V_{SC}$ and the parallel magnetic field $B_\parallel$. In such spectra, the nonlocal conductance is finite only in an energy window between $\Delta_{SC}$ and $\Delta_i$. For a given trace of the nonlocal conductance as a function of bias voltage, we determine an adaptive threshold based on the noise level at a large bias voltages. $\Delta_{SC}$ and $\Delta_i$ are estimated by checking when the nonlocal conductance exceeds the threshold value. This is done independently for both $g_{RL}$ and $g_{LR}$ as well as both positive and negative bias values. This results in four estimates of $\Delta_{SC}$ and $\Delta_i$ each, from which the mean and standard deviation are calculated and presented in the figures. Values of the nonlocal slope $S_{RL}$ and $S_{LR}$ are estimated as the numerical derivative of the data at zero bias voltage, after application of a Savitzky-Golay filter. A detailed description and examples can be found in the supplementary information.

## Data availability
The raw data generated in this study, as well as the code used to analyze the data, have been deposited in the repository on Zenodo and are available at https://doi.org/10.5281/zenodo.6913897[41].

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

## Acknowledgements

The authors thank Sebastian Heedt, Andrey Antipov, Dmitry Pikulin, Bernard van Heck, Michael Wimmer, Anton Akhmerov, Leo Bourdet and Georg W. Winkler for helpful discussions. We also thank Jan Cornelis Wolff, Mark Ammerlaan, Olaf Benningshof and Jason Mensingh for valuable technical support. This work has been financially supported by the Dutch Organization for Scientific Research (NWO) and Microsoft Corporation Station Q.

## Author contributions

L.P.K., G.P.M., and N.v.L. conceived the experiment. G.P.M. and N.v.L. fabricated and measured the devices. J.Y.W., R.C.D. T.D, G.W., A.B., D.v.D, M.L., and C.S. assisted with sample fabrication and/or measurements. G.P.M. and N.v.L. analyzed the transport data. G.B., S.G. and E.P.A.M.B. carried out the nanowire synthesis. G.P.M. and N.v.L. wrote the manuscript with valuable input from all authors. L.P.K. supervised the project.

## Competing interests

The authors declare no competing interest.
