## [Peer review file · Nature Communications]

REVIEWER COMMENTS

Reviewer #1 (Remarks to the Author):

I have read with interest the manuscript by Loo, Mazur et al. The text is skillfully written, the presentation of the data is clear and concise, and overall the work is of high scientific quality. In the following I reply to the guiding questions one-by-one:

1) What are the noteworthy results?

The main claim of the manuscript is the measurement of the weak coupling regime for proximitized superconductivity in a semiconducting wire. This is achieved by depleting the wire of charge carriers at the superconductor-semiconductor interface, using a gate electrode. It is important to notice that the strong coupling regime in the same (or very similar) devices has already been published by the same authors this year in *Advanced Materials* (Ref. 22 in the manuscript). In Ref. 22 the authors explicitly mention holding back the gate dependence data to be published in a separate manuscript (i.e. the current one). From my point of view the gate dependence which gives the weak coupling regime presented here completes (or actually is part of) the scientific story presented in Ref. 22. This is the main reason why I can not recommend the manuscript for publication in a high-impact journal such as *Nature Communications*.

2) Will the work be of significance to the field and related fields? How does it compare to the established literature? If the work is not original, please provide relevant references.

In my opinion the current work will be of significance for the field of hybrid devices, but not much for related fields. When I say this I keep in mind that the results presented by the same authors in Ref. 22 will probably be of interest also for related fields.

3) Does the work support the conclusions and claims?

Yes. The experimental evidence is clear, and in my opinion the measured effect is beyond doubt.

4) Are there any flaws in the data analysis, interpretation and conclusions?

I do not see any flaws in the data analysis. The work is carried out very professionally.

5) Is the methodology sound? Does the work meet the expected standards in your field?

Yes. The methodology follows the standards in the community. I do not see controversial results. The effect measured and reported here is expected following the results of Ref. 22.

6) Is there enough detail provided in the methods for the work to be reproduced?

While reproducibility is generally a problem for these devices and for the community, the methods used by the authors are clear.

Reviewer #2 (Remarks to the Author):

The paper is well-written and figures are clear. The authors measure the induced gap and parent gap of their hybrid devices using nonlocal spectroscopy in 3 NWs of lengths 640nm, 1 μ m and 8 μ m. They tune the strength of the superconductor-semiconductor coupling by pushing/pulling the electronic wavefunction towards/away from the superconducting shell of the NW using a gate voltage, but they also highlight that the change in coupling could be due to changing density and occupying additional subbands with weaker coupling and it seems it is not exactly clear what the mechanism is. The 1 μ m device is studied and shows the in-plane induced critical field increases with decreasing gate voltage ("stronger coupling") and shows a closing and reopening of the induced gap at intermediate coupling but without ZBP in the local spectra (closing/reopening was only observed in 1 of 11 total devices). They offer a proposal that topology is a possible explanation if the MZMs are pushed away from the leads and towards the middle of the NW it is not clear 1 μ m device is long enough for the MZMs to separate so the topological picture is not conclusive to me. Authors also offer other competing nontopological explanations as well. While I find the paper instructive I am not convinced it possess the novelty for publication in Nature communications.

Reviewer #3 (Remarks to the Author):

Summary:

In "Electrostatic control of the proximity effect in the bulk of semiconductor-superconductor hybrids" the authors study the tunability of the proximity effect of a semiconductor-superconductor nanowire hybrid through electrostatic gating. More specifically, they demonstrate the crossover between strong and weak coupling regimes by changing the voltage of the gate underneath the proximitized region and measuring the parent gap of the superconductor and the induced, bulk gap of the hybrid using non-local tunnel spectroscopy in a three terminal geometry. Essentially, transport from one normal lead to the other is only possible within the energy window between the induced gap and the gap of the parent superconductor, which allows the author to extract these values from their non-local transport measurements.

This demonstration of the crossover between the strong and weak coupling regimes is an important step towards the reliable realization of Majorana zero modes in semiconductor-superconductor hybrids since the probability of realizing the topological phase with well separated Majorana zero modes is dramatically reduced if the device is too deep into either the strong or weak coupling regimes. For this reason, I find the current paper a valuable contribution to the field. Overall, the methods and interpretations presented in the paper seem valid. In addition, the text is quite accessible and clear except for a few places (see below).

Questions/Issues:

1. In the introduction, the authors state “Nonlocal transport is possible only in an energy window between the gap of the superconductor and the induced gap in the semiconductor, and thus can be used to directly determine the induced gap in the bulk of the hybrid.” While I agree with the basic message of this sentence, it’s technically incorrect since the non-local transport can only detect delocalized states that couple to both leads. Localized states in the bulk of the wire may have an induced gap smaller than what is detected in the non-local transport, so strictly speaking the experiment is measuring the induced gap of delocalized states. This is perfectly fine, but I think it should be clarified in the text.

2. On page 3, the authors state “By tuning the super gate voltage, the induced gap can be reduced to roughly half of its initial value which implies that all semiconductor states that couple to both leads reside above an energy threshold even at strong electric fields. We suspect this is the result of the finite size of the nanowire, where the potential barriers at the ends provide a source of mixing between proximitized and unproximitized states in the hybrid segment.” I believe this interpretation is incorrect. The induced gap is not diminished with increasing supergate voltage $V_{\{SG\}}$ due to some increased mixing with the states of the unproximitized region. Rather, it is due to wavefunction in the cross section of the nanowire shifting away from the semiconductor-superconductor interface as illustrated in Fig. 1(b).

3. On page 3, the authors state “the induced gap can be fully closed for long nanowires” (at zero magnetic field), but give no physical explanation of this observation. What the reason the induced gap at zero magnetic field can close for long wires but not short wires? If the induced gap is controlled by the wavefunction profile in the cross section of the nanowire, presumably the induced gap should be essentially independent of the hybrid length. A plausible interpretation to me that in a longer wire, device inhomogeneity has a greater chance of creating a region of reduced coupling between the semiconductor and superconductor.

4. In discussing possible explanations for the closing and reopening of the gap without the presence of zero-bias peaks, the authors bring up the Little-Parks effect as a possible explanation, stating that “It has been shown theoretically that such effects can happen in the geometry used in this work.” While I understand that the exact effective flux is undetermined due to the wavefunction profile in the cross section of the wire being unknown, it seems to me like one could estimate what the effective area would have to be in order for the Little-Parks effect to be a candidate explanation since a flux quantum has to be threaded through the wavefunction. It may help the authors even rule this out as a possibility.

5. In discussing possible explanations for the closing and reopening of the gap without the presence of zero-bias peaks, the authors state “Yet, it may be possible that the presence of tunnel gates generates a smooth potential profile near the ends of the wire. In this case, the local spectra only represent the presence of bound states formed on the smooth potential, while pushing the Majorana zero modes towards the center of the hybrid - effectively decoupling them from the leads.” I agree that this is a possibility. I think it’s important, however, to point out that the tunnel gate potential is not the only possibility. Device disorder independent of the tunnel gates can also do this.

6. As a suggestion to improve the work, there is little connection made to the local conductance except to say that zero-bias peaks are not found in the local conductance with the closing and reopening of the gap. In the limit of a clean device, there is a one-to-one correspondence between the non-local and local conductance since there are no localized states (except Majorana zero modes in the topological phase). While I understand that the focus of this work is to explore the tunability of the bulk properties of the hybrid’s delocalized states, a comparison of the local and non-local conductance may be useful as it gives a sense of how many localized states are present.

7. Unless I missed it, I don’t think dimensions of the cross section of the nanowire are given. This may be useful to include, since the dimensions of the nanowire greatly affect the electrostatics and subband occupation.

8. In supplementary pg. 8, the authors state “We suspect that some of these states can acquire a finite charging energy as their wave function is pulled away from the semiconductor-superconductor interface due to strongly positive super gate voltages. This can allow them to cross zero energy, resembling the closing of the induced gap.” Why does having a finite charging energy allow for a zero energy crossing? If this is explained in some paper, there should be a reference. Why are these zero-energy crossing not consistent with just having a finite broadening from the leads?

We would like to thank all the referees for the insightful comments regarding our paper. We have modified the paper accordingly. Please find our point-by-point response below. Our answers are marked in blue and modifications to the paper are marked in red. Likewise, adjustments in the paper's main text are marked with red.

On behalf of all authors:

N. van Loo
G.P. Mazur
L.P. Kouwenhoven

Reviewer #1 (Remarks to the Author):

I have read with interest the manuscript by Loo, Mazur et al. The text is skillfully written, the presentation of the data is clear and concise, and overall the work is of high scientific quality. In the following I reply to the guiding questions one-by-one:

We thank the referee for the positive words about our work.

1) What are the noteworthy results?

The main claim of the manuscript is the measurement of the weak coupling regime for proximitized superconductivity in a semiconducting wire. This is achieved by depleting the wire of charge carriers at the superconductor-semiconductor interface, using a gate electrode. It is important to notice that the strong coupling regime in the same (or very similar) devices has already been published by the same authors this year in *Advanced Materials* (Ref. 22 in the manuscript). In Ref. 22 the authors explicitly mention holding back the gate dependence data to be published in a separate manuscript (i.e. the current one). From my point of view the gate dependence which gives the weak coupling regime presented here completes (or actually is part of) the scientific story presented in Ref. 22. This is the main reason why I can not recommend the manuscript for publication in a high-impact journal such as *Nature Communications*.

As noticed by the referee, the current work is in line with the series of works aimed at improving fabrication technology and material properties of InSb based semiconductor-superconductor hybrids. Ref 22. specifically deals with the problem of improving B-field compatibility of Al based hybrids, which we regard as a complete scientific story. In our opinion the present manuscript deals with fundamentally different physics, albeit using similar devices. While both manuscripts look at the superconducting gap, here we use its gate tunability to study the semiconductor-superconductor coupling. The conclusions presented in our manuscript are valid for pure Al as well, and in fact for any superconductor, where the fine details will be ultimately determined by the band offset at the semiconductor/metal interface. It is precisely the understanding of the gate-tunable coupling which forms the foundation

for next-generation experiments, such as the realization of a Kitaev chain. We elaborate on this and would like to emphasize the general importance of reported results below:

1) Relevance for the search of topological superconductivity based on the Lutchyn-Oreg model.

In the original works by Lutchyn and Oreg, superconductivity was simply assumed in the semiconducting wire by means of a phenomenological pairing parameter. In contrast, Antipov et al. (Effects of Gate-Induced Electric Fields on Semiconductor Majorana Nanowires | Phys. Rev. X.) provided a paradigm shift on the theory side by showing that a microscopic treatment is necessary to understand proximitized nanowires. Our work provides the experimental counterpart to the theory work from Antipov et al. Our data provides valuable input to simulations such as these, which can be used to extract valuable information like the band offset between the materials. This on its own is important, because it determines whether a topological phase can be realized at all given a particular material combination - thus, we believe any future candidate materials (like Tin or Lead on InSb, InAs or PbTe) should go through the same experiments we demonstrated here.

Only two experiments (de Moor et al., Vaitiekėnas et al.) have tried to study the gate response of the proximity effect in the past. However, neither of them have done so systematically and both relied on local spectroscopy. Yet, the effects of the semiconducting tunnel junctions (such as smooth potential, disorder etc.) have largely obscured the real properties of proximitized semiconductors in the last decade of Majorana research. Our work is the first to overcome the inadequacies of previous works (i.e. the use of local spectroscopy), by using nonlocal spectroscopy and, for the first time, mapping the gate voltage - magnetic field phase diagrams for proximitized hybrids. We also like to stress that all the specimens studied in this work exhibit qualitatively the same behavior which is a major technological step forward given reproducibility issues pointed out by the referee.

2) Relevance for the experiments beyond Lutchyn-Oreg's model.

The ability to gate proximitized semiconductors underneath the metal is still surprising to some, yet our work explicitly demonstrates that one can combine the gate-tunability of a semiconductor with the pairing provided by the superconductor.

Our results apply to any system regarding the semiconductor- superconductor proximity effect: the tunable coupling determines where in the gate space one needs to go to obtain the favorable properties of both, without sacrificing the properties of the other. With the present study, we are establishing that a large part of the parameter space in the gate chemical potential is irrelevant, as the wire is either strongly metallized or too weakly proximitized. We believe that it is a very important message to the community as it gives fundamental understanding of how such hybrids work. This knowledge is now already applied to realize major-breakthrough experiments, such as demonstration of equal-spin crossed Andreev reflection [Wang, Dvir, Mazur et al., Nature, 612, 448-453] and the formation of a long-sought-after Kitaev chain [Dvir, Wang, van Loo et al., arXiv:2206.08045]. These experiments are operated precisely at the crossover between the weak and strong coupling regimes, where the hybrids both have appreciable spin-orbit interaction and g-factor, a moderate electron density and a hard induced superconducting gap.

3) Reopening of the superconducting gap

We would like to underline that we report on the reopening of the induced superconducting gap, when the device is tuned to the crossover between strongly and weakly coupled regimes. Observation of a gap reopening is still one of the key signatures of topological superconductivity and so far it has been reported scarcely. More importantly, we provide alternative scenarios which do not involve a topological phase transition. This is a perspective we believe will strongly benefit the research field, which is plagued by strong claims that are not backed up by the data. Indeed, we show for the first time that the observation of a gap reopening is not necessarily related to a topological phase transition.

4) Control and understanding of the soft gap.

In addition we have modified Figure 2 of the manuscript such that it clearly illustrates behavior of the gap with respect to the semiconductor/superconductor coupling. We also stress that a long-standing problem in the field - i.e. the “soft” induced superconducting gap can be achieved by sufficiently decoupling the hybrid wavefunction from the superconductor, where in many previous studies was just a consequence of poor transmission at the interface.

2) Will the work be of significance to the field and related fields? How does it compare to the established literature? If the work is not original, please provide relevant references.

In my opinion the current work will be of significance for the field of hybrid devices, but not much for related fields. When I say this I keep in mind that the results

presented by the same authors in Ref. 22 will probably be of interest also for related fields.

3) Does the work support the conclusions and claims?

Yes. The experimental evidence is clear, and in my opinion the measured effect is beyond doubt.

4) Are there any flaws in the data analysis, interpretation and conclusions?

I do not see any flaws in the data analysis. The work is carried out very professionally.

5) Is the methodology sound? Does the work meet the expected standards in your field?

Yes. The methodology follows the standards in the community. I do not see controversial results. The effect measured and reported here is expected following the results of Ref. 22.

6) Is there enough detail provided in the methods for the work to be reproduced?

While reproducibility is generally a problem for these devices and for the community, the methods used by the authors are clear.

Reviewer #2 (Remarks to the Author):

The paper is well-written and figures are clear. The authors measure the induced gap and parent gap of their hybrid devices using nonlocal spectroscopy in 3 NWs of lengths 640nm, 1 μ m and 8 μ m. They tune the strength of the superconductor-semiconductor coupling by pushing/pulling the electronic wavefunction towards/away from the superconducting shell of the NW using a gate voltage, but they also highlight that the change in coupling could be due to changing density and occupying additional subbands with weaker coupling and it seems it is not exactly clear what the mechanism is. The 1 μ m device is studied and shows the in-plane induced critical field increases with decreasing gate voltage ("stronger coupling") and shows a closing and reopening of the induced gap at intermediate coupling but without ZBP in the local spectra (closing/reopening was only observed in 1 of 11 total devices).

They offer a proposal that topology is a possible explanation if the MZMs are pushed away from

the leads and towards the middle of the NW it is not clear 1 μ m device is long enough for the MZMs to separate so the topological picture is not conclusive to me. Authors also offer other competing nontopological explanations as well. While I find the paper instructive I am not convinced it possess the novelty for publication in Nature communications.

We thank the referee for careful reading of the manuscript.

We have modified Fig 2. and Fig 3. to clarify on the point we're trying to make. Fig 2. presents the data only from the long (8 micrometer) hybrid nanowire. Now the figure shows the full conductance matrix measured at zero magnetic field as a function of

super gate voltage. We observe a clear reduction of semiconductor/superconductor coupling which in the phenomenological picture can be viewed as pulling the wave function of the hybrid towards the nanowire.

We are still warning the reader that the exact microscopic picture remains unclear. In particular, how many subbands and how strongly are they coupled to the superconductor are out of reach for the current state-of-the-art experimental techniques. Yet, the effects are theoretically predicted and related to each other, which we clarify in the following paragraph:

However, the application of an electric field does not exclusively tune the coupling but also controls the density in the hybrid. Typically, we observe a sudden onset of the reduction of Δ_i while the magnitude of the nonlocal signal increases concurrently. This behavior has theoretically been related to the occupation of an additional subband with a reduced coupling [8]. Still, it remains unknown how many sub-bands are active in our hybrids.

Regarding the interpretation of the observed reopening, we want to clarify that none of the presented pictures is favored by the authors. We agree with the referee, that the hybrid is likely within the short wire limit. This is explicitly said in the text:

On the contrary, it is also possible that the reopening of the gap has a topologically trivial origin. For example, the hybrid segment is only 1 μm long such that it can be within the short wire limit. This results in a spectrum comprising of discrete energy levels with a small energy spacing, making the concept of topology ill-defined [29]

However, as the microscopic details such as coherence length remain unknown for the hybrid wires, we are discussing different scenarios which could potentially explain the gap reopening.

As Referee #1 also shared concerns about the novelty of the manuscript, we refer Referee #2 to the answer above.

Reviewer #3 (Remarks to the Author):

Summary:

In “Electrostatic control of the proximity effect in the bulk of semiconductor-superconductor hybrids” the authors study the tunability of the proximity effect of a semiconductor-superconductor nanowire hybrid through electrostatic gating. More specifically, they demonstrate the crossover between strong and weak coupling

regimes by changing the voltage of the gate underneath the proximitized region and measuring the parent gap of the superconductor and the induced, bulk gap of the hybrid using non-local tunnel spectroscopy in a three terminal geometry. Essentially, transport from one normal lead to the other is only possible within the energy window between the induced gap and the gap of the parent superconductor, which allows the author to extract these values from their non-local transport measurements.

This demonstration of the crossover between the strong and weak coupling regimes is an important step towards the reliable realization of Majorana zero modes in semiconductor-superconductor hybrids since the probability of realizing the topological phase with well separated Majorana zero modes is dramatically reduced if the device is too deep into either the strong or weak coupling regimes. For this reason, I find the current paper a valuable contribution to the field. Overall, the methods and interpretations presented in the paper seem valid. In addition, the text is quite accessible and clear except for a few places (see below).

Questions/Issues:

1. In the introduction, the authors state “Nonlocal transport is possible only in an energy window between the gap of the superconductor and the induced gap in the semiconductor, and thus can be used to directly determine the induced gap in the bulk of the hybrid.” While I agree with the basic message of this sentence, it’s technically incorrect since the non-local transport can only detect delocalized states that couple to both leads. Localized states in the bulk of the wire may have an induced gap smaller than what is detected in the non-local transport, so strictly speaking the experiment is measuring the induced gap of delocalized states. This is perfectly fine, but I think it should be clarified in the text.

The referee is correct that technically, nonlocal transport is in principle carried via states which couple to both leads. We have adjusted the main text to be more precise:

Nonlocal transport is fundamentally carried by states in the nanowire that couple to both leads. Moreover, it requires their energy to reside in an energy window between the gap of the superconductor and the induced gap in the semiconductor [17], and thus can be used to directly determine the induced gap in the bulk of the hybrid [18].

We later on, when describing the methodology, alert the reader to caveats of nonlocal measurements, which we elaborate on in the supplemental information:

While this picture helps to understand three-terminal measurements, we note that nonlocal processes can involve energy relaxation of the injected electrons as well as non-equilibrium effects not captured by the single-particle transport theory [21]. We further elaborate on this in Supplementary section II.B.

Section II.B of the supplementary information reads as follows:

In this work, we use nonlocal spectroscopy to investigate the bulk properties of three-terminal InSb/Al/Pt nanowire hybrids. In particular, nonlocal transport is facilitated through the density of states between Δ_{SC} and Δ_i . However, several processes complicate this simple picture. For example, the visibility of nonlocal signals is affected strongly by the non-ideal injection and detection processes in the tunnel junctions [4]. Moreover, relaxation from above to below Δ_{SC} is sometimes visible in the nonlocal spectra [4]. In addition, Δ_{i} can potentially vary along the length of the hybrid. As a result, the nonlocal signal may reflect the largest induced gap somewhere in the bulk. However, nonlocal transport is likely insensitive to fluctuations of Δ_i on a short length scale as quasiparticles can cross such areas through various tunneling mechanisms. Despite the above-mentioned complications, we systematically observe a good correspondence between the two nonlocal signals g_{RL} and g_{LR} , which supports the assumption that nonlocal transport can be used to evaluate bulk properties.

2. On page 3, the authors state “By tuning the super gate voltage, the induced gap can be reduced to roughly half of its initial value which implies that all semiconductor states that couple to both leads reside above an energy threshold even at strong electric fields. We suspect this is the result of the finite size of the nanowire, where the potential barriers at the ends provide a source of mixing between proximitized and unproximitized states in the hybrid segment.” I believe this interpretation is incorrect. The induced gap is not diminished with increasing supergate voltage V_{SG} due to some increased mixing with the states of the unproximitized region. Rather, it is due to wavefunction in the cross section of the nanowire shifting away from the semiconductor-superconductor interface as illustrated in Fig. 1(b).

The referee is correct that the induced gap diminishes due to the shifting wavefunction away from the semi-super interface. This is the message we tried to convey, however our attempt to distinguish between short and long nanowires made the section confusing. We have moved the short nanowire data to the supplement and altered Fig. 2. to focus on the diminishing of the induced gap as a result of weakened semiconductor-superconductor coupling. We have also altered the corresponding text, which we believe is now more readable and less confusing. As it is quite a long section, we refer to the main text to see the changes we have made.

3. On page 3, the authors state “the induced gap can be fully closed for long nanowires” (at zero magnetic field), but give no physical explanation of this observation. What the reason the induced gap at zero magnetic field can close for long wires but not short wires? If the induced gap is controlled by the wavefunction

profile in the cross section of the nanowire, presumably the induced gap should be essentially independent of the hybrid length. A plausible interpretation to me that in a longer wire, device inhomogeneity has a greater chance of creating a region of reduced coupling between the semiconductor and superconductor.

We thank the referee for this helpful comment which helps to make our work more clear. The message we tried to convey is that nanowires in the weak-coupling regime may have unproximitized states in the hybrid segment, which causes the induced gap to close. However, such states can obtain a finite energy gap through mixing with proximitized states, also residing in the hybrid segment. The reason we believe this happens only in short wires is that the tunnel barriers themselves act as strong scatterers which mix these states. We have moved the discussion about short and long wires to the supplement, as it seems to distract from the main message. We have added a paragraph in the supplemental information page 9 to describe this effect:

We note that the induced gap in short hybrids does not close at zero magnetic field, as can be seen for example in Fig. S7. This is a consequence of the short length of this hybrid: while states without any semiconductor-superconductor coupling may form in the weak-coupling regime, in practice these can obtain a finite energy gap if they are allowed to mix with proximitized states. Such mixing can occur due to disorder and, in this particular case of short hybrids, due to the presence of tunnel junctions at the ends of the hybrid segment.

4. In discussing possible explanations for the closing and reopening of the gap without the presence of zero-bias peaks, the authors bring up the Little-Parks effect as a possible explanation, stating that “It has been shown theoretically that such effects can happen in the geometry used in this work.” While I understand that the exact effective flux is undetermined due to the wavefunction profile in the cross section of the wire being unknown, it seems to me like one could estimate what the effective area would have to be in order for the Little-Parks effect to be a candidate explanation since a flux quantum has to be threaded through the wavefunction. It may help the authors even rule this out as a possibility.

Based on the closing and reopening fields, we estimate a wavefunction cross section diameter around 20-30 nm. This is significantly smaller than the nanowire diameter of ~126 nm. However, the little-parks effect relies on interference and thus requires a ring-like structure. We think it is unlikely that the wavefunction forms a ring-like geometry together with the superconductor given the estimated diameter. We initially attempted to list all possibilities, however we think this can exclude such effects thanks to the referee’s suggestion. We have accordingly removed this part from the manuscript.

5. In discussing possible explanations for the closing and reopening of the gap without the presence of zero-bias peaks, the authors state “Yet, it may be possible that the presence of tunnel gates generates a smooth potential profile near the ends of the wire. In this case, the local spectra only represent the presence of bound states formed on the smooth potential, while pushing the Majorana zero modes towards the center of the hybrid - effectively decoupling them from the leads.” I agree that this is a possibility. I think it's important, however, to point out that the tunnel gate potential is not the only possibility. Device disorder independent of the tunnel gates can also do this.

We agree with the referee that disorder can result in the same effect. We have adjusted the main text to accommodate this:

Similar effects are expected to be caused by the device disorder independent of the tunnel gate voltage [31].

6. As a suggestion to improve the work, there is little connection made to the local conductance except to say that zero-bias peaks are not found in the local conductance with the closing and reopening of the gap. In the limit of a clean device, there is a one-to-one correspondence between the non-local and local conductance since there are no localized states (except Majorana zero modes in the topological phase). While I understand that the focus of this work is to explore the tunability of the bulk properties of the hybrid's delocalized states, a comparison of the local and non-local conductance may be useful as it gives a sense of how many localized states are present.

Following the referee's suggestion we modified Fig.2 and Fig.3 such that the full conductance matrix is visible to the reader. Indeed, it is clearly visible that in the strong-coupling regime there are localized states in the local conductance spectra which are otherwise absent in non-local conductance spectra. We believe that inclusion of the local conductance spectra makes the work more complete. We agree that the local spectra do give a sense on the number of localized states near the junctions and help to appreciate measurements of the full conductance matrix. We have adjusted the main text to accommodate the changed figures, and mention the relation between local and nonlocal signals explicitly in the main text:

However, some states can be seen in these spectra which do not correlate between the two panels nor show up in the nonlocal signals - a confirmation that these states are confined to the local tunnel junctions.

and in the conclusion:

Even though this has been attempted in the past using two-terminal experiments [10, 11], local tunneling spectroscopy does not allow for discriminating between states localized in the junction area which are known to possess a gate-tunable coupling

[37]. Such states are present in virtually all our local spectra, which demonstrates that nonlocal measurements are truly necessary to properly investigate the bulk properties of semiconductor-superconductor hybrids.

7. Unless I missed it, I don't think dimensions of the cross section of the nanowire are given. This may be useful to include, since the dimensions of the nanowire greatly affect the electrostatics and subband occupation.

We agree that the nanowire cross-sections are a relevant piece of information. We have explicitly added the dimensions of the nanowires to the supplemental figure S6, where we show the overview of measured devices. In particular, the two devices presented in the main text (Device B and C) are of similar diameter ~120nm:

Device A - length: 640 nm, diameter: 79 nm, Device B - length: 8 μm , diameter: 120 nm, Device C - length: 1 μm , diameter: 126 nm, Device D - length: 240 nm, diameter: 75 nm, Device E - length: 450 nm, diameter: 77 nm, Device F - length: 840 nm, diameter: 81 nm

8. In supplementary pg. 8, the authors state "We suspect that some of these states can acquire a finite charging energy as their wave function is pulled away from the semiconductor-superconductor interface due to strongly positive super gate voltages. This can allow them to cross zero energy, resembling the closing of the induced gap." Why does having a finite charging energy allow for a zero energy crossing? If this is explained in some paper, there should be a reference. Why are these zero-energy crossing not consistent with just having a finite broadening from the leads?

In the limit of short nanowires, one can model the hybrid as a proximitized quantum dot following Pillet et al., *Andreev bound states in supercurrent-carrying carbon nanotubes revealed* (Andreev bound states in supercurrent-carrying carbon nanotubes revealed | Nature Physics), Nature Physics 2010. The energy of ABSs formed in such a dot can be controlled via the chemical potential and otherwise rely on the coupling to the superconductor and the charging energy. In this model, the ABS energy can only cross zero energy in the presence of a charging energy, which we think is the case for these particular devices. We have added the citation to the supplementary section.

REVIEWERS' COMMENTS

Reviewer #1 (Remarks to the Author):

I thank the authors for their detailed reply. Their rationale is now more clear and I don't have strong objections to raise against the publication of this manuscript. It constitutes a complex and technically challenging study which completes Ref 24 (former ref. 22).

Reviewer #3 (Remarks to the Author):

I am satisfied with the responses to my questions/concerns. I believe the paper has improved with focusing on the long hybrid device (Device B - 8 micron), avoiding the confusion about the role of tunnel barriers in long vs short wires. I also think that including the local conductance results has improved the work. It gives a more complete description of the device.

I disagree with Referee 1's opinion that "the gate

dependence which gives the weak coupling regime presented here completes (or actually is part of) the scientific story presented in Ref. 22." I agree with the authors that Ref. 22 (Ref. 24 in the resubmitted paper) is a separate story from this work. Ref. 22's main focus is on improving the robustness of superconductivity in the parent superconductor to magnetic fields. This can be studied exclusively in the strong-coupling regime since it's essentially independent the semiconductor. In the current work, the focus is on tuning between the strong and weak coupling regimes. This is an orthogonal issue to the robustness of superconductivity in the parent superconductor. After all, tuning between the strong and weak coupling regimes can be studied at zero magnetic field (see Fig. 2 of the current work).